# *In vitro* experimental conditions and tools can influence the safety and biocompatibility results of antimicrobial electrospun biomaterials for wound healing

**Kaisa Põhako-Palu[1], Kairi Lorenz[1], Kelli Randmäe[1], Marta Putrinš[1], Külli Kingo[2,3], Tanel Tenson[4], Karin Kogermann[1]** *

**1** Institute of Pharmacy, University of Tartu, Tartu, Estonia, **2** Dermatology Clinic, Tartu University Hospital, Tartu, Estonia, **3** Institute of Clinical Medicine, University of Tartu, Tartu, Estonia, **4** Institute of Technology, University of Tartu, Tartu, Estonia

* karin.kogermann@ut.ee

**Data Availability Statement:** All relevant data are within the manuscript and its Supporting Information files.

## Abstract

Electrospun (ES) fibrous nanomaterials have been widely investigated as novel biomaterials. These biomaterials have to be safe and biocompatible; hence, they need to be tested for cytotoxicity before being administered to patients. The aim of this study was to develop a suitable and biorelevant *in vitro* cytotoxicity assay for ES biomaterials (e.g. wound dressings). We compared different *in vitro* cytotoxicity assays, and our model wound dressing was made from polycaprolactone and polyethylene oxide and contained chloramphenicol as the active pharmaceutical ingredient. Baby Hamster Kidney cells (BHK-21), human primary fibroblasts and MTS assays together with real-time cell analysis were selected. The extract exposure and direct contact safety evaluation setups were tested together with microscopic techniques. We found that while extract exposure assays are suitable for the initial testing, the biocompatibility of the biomaterial is revealed in *in vitro* direct contact assays where cell interactions with the ES wound dressing are evaluated. We observed significant differences in the experimental outcome, caused by the experimental set up modification such as cell line choice, cell medium and controls used, conducting the phosphate buffer washing step or not. A more detailed technical protocol for the *in vitro* cytotoxicity assessment of ES wound dressings was developed.

## Introduction

Currently, there are various novel biomaterials under development that can be applied in different medical fields due to their superior properties and functionality compared to the conventional materials. Some of the examples of biomaterials are implants, heart valves, stents, lenses, prostheses, wound dressings etc [1]. Biomaterials are widely used in regenerative medicine, tissue engineering, orthodontics and drug delivery [2]. The terms and definitions of "biomaterials" have changed a lot over the past decades, and nowadays artificial organs, tissue

**Funding:** Estonian Research Council grant number PRG1507 (Karin Kogermann).

scaffolds and delivery vehicles for cells, genes and active pharmaceutical ingredients can all be classified as biomaterials [3]. Although more commonly these drug-loaded biomaterials are termed as drug delivery systems (DDSs). Biomaterial is an engineered material/substance that has been engineered to become in contact with living tissues for therapeutic or diagnostic purposes. The latter also defines the exact properties that the biomaterial needs to have since every biomaterial should be prepared for a specific purpose [4–6].

Electrospun (ES) materials can be classified as biomaterials when used as tissue scaffolds, wound dressings and DDSs [7]. ES mats have several prominent features, such as a porous structure, the abilities to incorporate drugs, including antibiotics, into fiber structures, and a high surface area to volume ratio for efficient drug release [8]. These features make ES fibers desirable delivery vehicles for the local therapy of skin and wounds. With antimicrobial agent-loaded ES wound mats, it is possible to reduce or avoid systemic antibiotic treatment and provide local antimicrobial treatment at higher drug concentrations [9]. ES wound mats may also accelerate wound healing because of their fibrous and porous structure, providing enough oxygen and moisture to the wound and a supportive surface for eukaryotic cells to attach and grow, and are used as dressings or skin scaffolds, respectively [10]. More importantly, the presence of antimicrobial agent in ES wound mats is relevant as this inhibits the growth of bacteria which is also more favoured in these mentioned environmental conditions developed.

Biomaterials, including ES wound dressings, are characterized using various analytical methods that enable us to understand their properties (e.g., morphology, mechanical properties, degradation behaviour) and functionality (e.g., antimicrobial efficacy). Consequently, every biomaterial may be designed differently in line with its final application and should be characterized according to its purpose and place of use. For example, before applying new wound dressings to patients, it is crucial to test their safety, efficacy, and biocompatibility. Unfortunately, there is no "one test fits all" evaluation method suitable for different medical devices because of the variability in the body environment and interaction between the device and cells [11]. The International Organization for Standardization guidelines (ISO 10993–5:2009) suggest different methods suitable for cytotoxicity testing, but it has been stated that the suggested methods might not be explicit enough to draw conclusions about medical device safety [12].

Cytotoxicity tests are divided into three main groups: extract exposure, indirect and direct contact methods [13]. Extract exposure evaluation is the simplest cytotoxicity assay that uses extracts made from tested materials. This method is more suitable for materials that release soluble substances into cell culture media, whereas direct contact methods are more sensitive because cells come into contact with the test material [11]. The indirect contact method is in between the two, but mainly includes molecular filtration and agar overlay methods, and relies on the diffusion of toxic substances from the medical device [11]. Since no standardized methods are available and only broad instructions are given in guidelines, different tests have been used for the characterization of ES biomaterials, which is most likely one of the reasons why the evaluation processes for biomaterials are often complicated and market entry is delayed [3]. Such variability in testing setups and conditions makes the comparison between different reports on ES wound dressings impossible. After an exhaustive literature search, we found that both extract exposure and direct contact methods have been widely used, but not always together with microscopy, which would give better understanding about cell morphological changes when in contact with material. It has been shown that fluorescent staining together with microscopy enables to enlighten the structure of cells and depending on the selected fluorophore and its sensitivity and selectivity [14]. A recent report has shown the importance of complementary analytical methods for the characterisation of ES materials [15]. Furthermore, different cell lines have been used for cytotoxicity testing, mainly in mouse fibroblasts and human skin fibroblasts [9, 15–19]. When using different methods and cells, the exact

technical procedures during testing may affect the results of the experiments and their inter-pretation and are important to be explored in more detail. It is often neglected that the biologi-cal responses of the cells take place during testing, and cells respond to the environment and change their behaviour which might not be suitable for the selected assay. There are various assays based on various cell functions, such as cell adherence, cell membrane permeability, enzyme activity, ATP production, and co-enzyme production [20].

We have previously published the development and full physicochemical characterisation of PCL/PEO and PCL/PEO/CAM ES fibers [18], and investigated the antimicrobial effect of the antibiotic-loaded fibers against various wound pathogens and using different antimicrobial assays [18, 21, 22]. In the present study, we tested these ES materials in two different cell lines using indirect extract exposure and direct contact cytotoxicity evaluations. The aim was to observe how the use of different cell lines and modification of the experimental design would affect the results of the cytotoxicity measurements. In the direct contact evaluation, we varied the experimental design. In one part of the experiment, cells were seeded onto the ES fiber material, washed with PBS, placed into a new well plate, and the cell culture medium was changed. In another part, no PBS wash or medium replacement was performed. The results of the cytotoxicity assay were complemented by confocal microscopy together with DAPI and Alexa 568 staining in order to illustrate the cell morphology.

## Results

### Electrospun fiber materials and cell-lines used for the validation of cytotoxicity assays

Electrospinning of the tested materials was performed as previously described by Preem *et al.* [18]. Fiber diameter range and morphology were examined using SEM analysis. SEM images and size analysis revealed a wide variation in the fiber diameters throughout the mats (Fig 1A and 1B). The diameter of the fibers was in the range between 0.1 μm and 8 μm, where most of the fibers where in the range between 0.1 μm and 0.5 μm for both PCL/PEO and PLC/PEO/CAM fiber mats. The histograms of fiber size distribution illustrate the fibers obtained much more compared to the arithmetic mean diameter of the fibers. The structure of the obtained ES fiber mats resembles the skin´s extracellular matrix structure with heterogeneous fiber sizes. However, the diameter range and morphology of the ES fibers were less homogeneous, and the prepared fibers were thinner than those previously described by Preem *et al.* [18]. This is most probably due to the different batches of polymer used in the electrospinning process obtained from the polymer manufacturer, which shows variations in the molecular weight of the polymer.

The CA filter material was used as a nontoxic control material (Fig 1C). Two different eukaryotic cell lines were used for safety evaluation, namely BHK-21 and more biorelevant human skin primary fibroblast (PF) cells obtained from the patient (Fig 1D and 1E). As can be seen, these cells exhibit different sizes and morphologies, and hence provide more diverse information about the cell-fiber mat behaviour that was investigated in the present study. BHK-21 cells are smaller in size, whereas PF cells are much larger, covering up to 10 times larger substrate surface area than BFK-21 cells. Importantly, in all subsequent experiments, both cell lines were seeded at the same cell numbers per surface area of the substrate.

### Extract exposure assays—indirect impact of ES fibers and their released components to cell viability

The easiest way to test cytotoxicity caused by the material according to the ISO 10993–5:2009 standard is to make a material extract [13], which is also widely used to investigate ES fiber

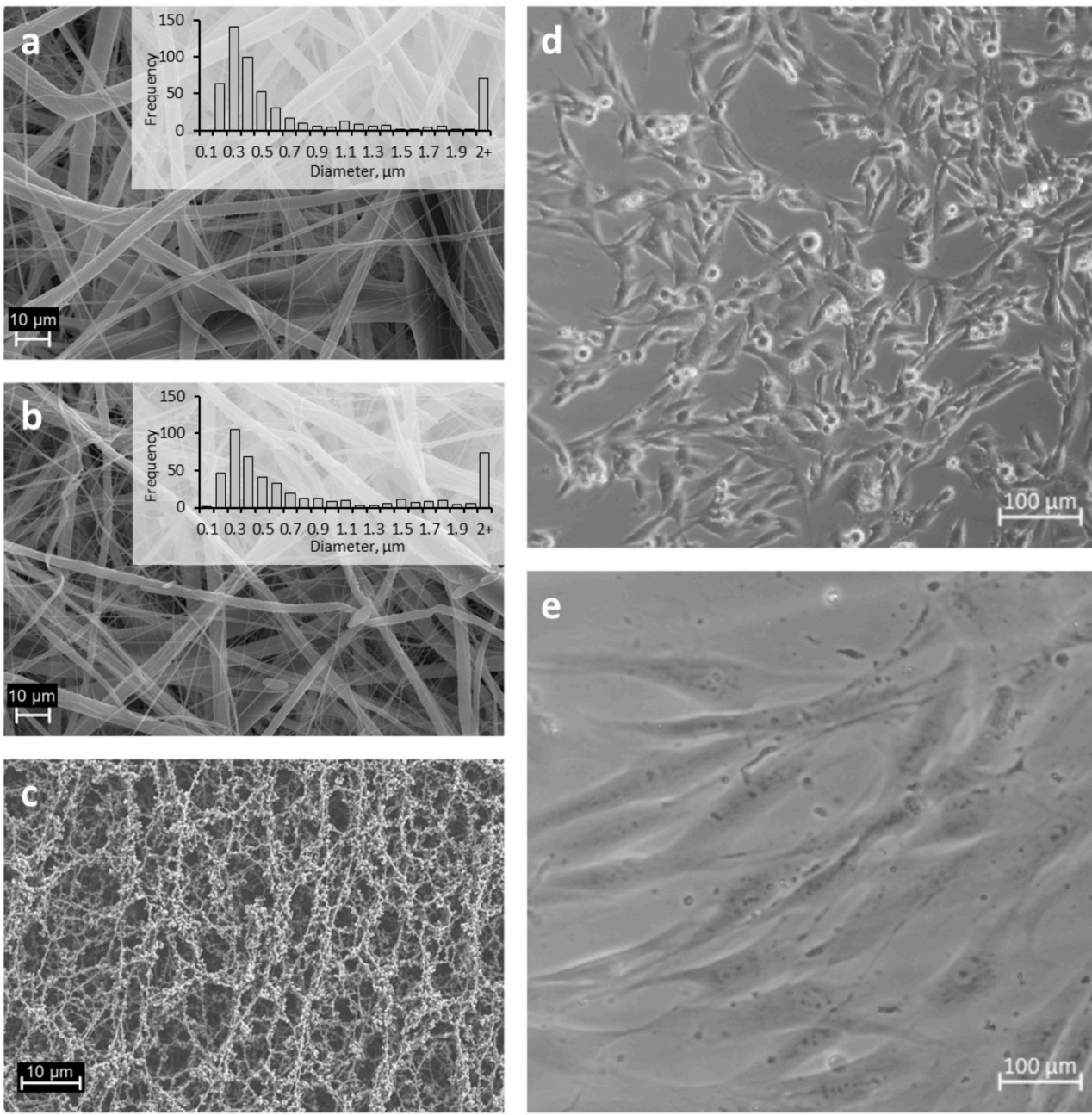

**Fig 1.** Scanning electron microscopy (SEM) images together with fiber diameter size distribution histograms of electrospun (ES) fiber mats made from pristine PCL/PEO (a) and PCL/PEO/CAM containing the antibacterial drug chloramphenicol (CAM) (b). Distribution of the fibers was measured manually, and the numbers of measured fibers were N = 544 for PCL/PEO and N = 488 for PCL/PEO/CAM. SEM image of the cellulose acetate (CA) filter material (c). Optical microscopy images of baby hamster kidney (BHK-21) cells (d) and human skin primary fibroblast (PF) cells (e).

material cytotoxicity [19, 23–25]. This means that the material is usually immersed in the cell medium and incubated for a specified period. For the measurement, the samples were removed and only the liquid medium (extract) was used for further analysis using different methods. Hence, the effect was evaluated indirectly. To evaluate cytotoxicity, the ISO 10993–5:2009 standard recommends the use of MTT based assays [13]. We previously investigated

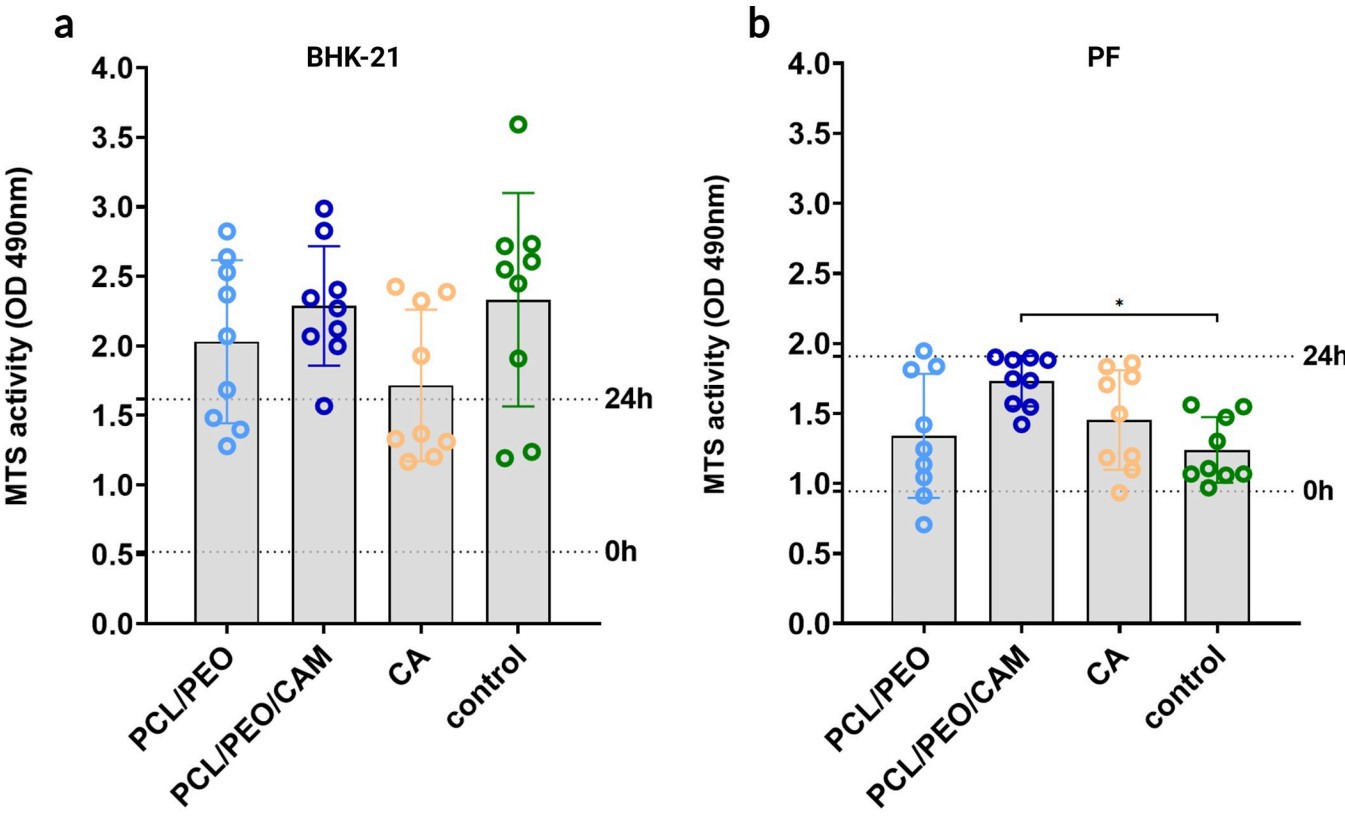

**Fig 2.** Results of the indirect extract exposure method using MTS assay and (a) BHK-21 and (b) PF cell lines. Key: 0 h control—MTS activity measured immediately after seeding; 24 h control–MTS activity of cells that were grown on the well plate bottom for 24 h before adding extracts. PCL/PEO–fluid extract of the PCL/PEO ES fiber mat; PCL/PEO/CAM–fluid extract of the PCL/PEO/CAM ES fiber mat; CA—fluid extract of cellulose acetate filter; Control–MTS activity was measured from untreated cells growing on the well plate well bottom for 48 h. The data represent three biological replicates from three separate experiments and are shown as mean ± SD. * Statistically significant difference (P ≤0.05) between marked groups.

the indirect cytotoxicity of these electrospun fiber mats using NIH 3T3 mouse fibroblasts and the CellTiter-Glo Luminescent Cell Viability Assay [18].

In this study, we used the MTS assay instead of the MTT assay, since it measures the activity of the same cellular enzymes, but the measurement of reaction products is more convenient with fewer experimental steps and is more rapid than the MTT assay [26]. MTS assays enable the measurement of cell viability and tracking of changes in cell proliferation at the chosen time points. MTS assay results are generally related to the live cell count and its increase or decrease over time, presented as MTS activity [27]. For this measurement, the extracts obtained from two ES fiber mats, PCL/PEO and PCL/PEO/CAM, were tested for safety, as described in the Methods section. When we evaluated the effect of ES fiber mats indirectly with extract exposure on eukaryotic cells, using MTS activity measurements, no cytotoxic effect caused by the tested ES fiber and CA filter extracts was observed (Fig 2). This result indicates that no substances harmful to cells were released from the fibers into the cell medium. A CA filter was used as the second control because of the similarities in the material features compared to the ES fiber materials. Similarly, to the ES fiber extract, the CA filter extract did not provoke any toxic effects on cells nor reduced their viability. In addition, no difference was observed between antibiotic-containing and antibiotic-free ES fibers in either cell line. This result proves that the antibiotic CAM released from the ES fiber does not have any toxic effects on cells at the concentrations used.

Next, we used RTCA as another method to analyse the safety of fiber mat extracts. This is also an indirect method that uses extracts from the tested samples. RTCA enables continuous and non-invasive measurement of cell growth and proliferation over time, making it a perfect method for testing the cytotoxicity of different materials and chemicals. This method is widely used to test drug-induced toxicities in drug development [28].

The cell index number measured by RTCA is directly correlated with the area of the well bottom covered by the cells [28]. Therefore, it is important to consider both cell number and cell size when seeding cells onto an E-16 well plate. We clearly observed a difference in cell size between the two cell lines used in this study (Fig 1D and 1E). Because the seeding amount of the cells was the same in both cell lines in all the experiments, PF cells covered the E-16 plate and 24-wellplate well bottom more densely than BHK-21 cells.

During incubation and continuous RTCA measurements, both cell well-being (attachment) and proliferation (increase in surface area covered) were detected. When the cell index values begin to decrease over time, they are related to cell detachment from the E-16 well plate. First, we set up the assay conditions in a normal growth medium (Fig 3A and 3B). In the PF cell line, we observed a decrease in the number of cells after the start of the experiment, indicating that the bottom of the well was densely covered with cells at the beginning of the experiment, and further cell proliferation could not be detected. A much smaller area of the well bottom was covered with BHK-21 cells at the beginning of the experiment; therefore, the cell index continued to increase after 48 h of incubation, indicating cell proliferation.

In the experimental setup, we tested both ES fiber mat extracts and different CAM concentrations, similar to the amount of CAM released from the fiber. The latter was performed to mimic the actual situation achieved during wound treatment using ES fiber mats. The RTCA results confirmed the MTS experimental results, and no toxic effects were observed in the ES fiber extracts (Fig 3C and 3D) or the presence of CAM in the growth medium (Fig 3E and 3F). The decrease in the cell indices of PF cells was not caused by the ES fiber mat extract, as the decrease was visible in both the test and control groups (Fig 3D and 3F).

One aspect that needs to be considered in such an experimental setup is the nature and specificity of the cell line used. There was a clear difference in the cell index values when experiments were conducted with PF cells using two different passage numbers (Fig 3D and S1 File). The experiments were performed with PF cells with passage number 5 (Fig 3D) and 12 (S1 File). The cell index values were higher when using the cells with lower passage number (Fig 3D) and phenomenon was reversed when using the cells with higher passage number (S1 File), but the overall shape of the graph remained the same. The effect of passage number on the experimental results for primary cells has been described previously in the literature [29, 30]. The effect of passage number on BHK-21, an immortalised cell line, was not observed in the experiments.

## Direct assessment of ES fiber mat cytotoxicity and biocompatibility (suitability as growth substrate)

The indirect testing of material safety through material extracts does not always provide sufficient information about whether the material itself can be safely used in contact with live cells and tissues. Specifically, when biomaterials are used as biodegradable scaffolds (e.g., artificial skin for wound healing). Therefore, direct safety measurement methods were used. A typical method to directly test material safety is to grow cells in contact with a material or object and then perform a cytotoxicity assay [13]. We tested our ES fiber materials as described in ISO 10993–5:2009 with some experimental modifications, where cells were grown on top of the ES fiber material.

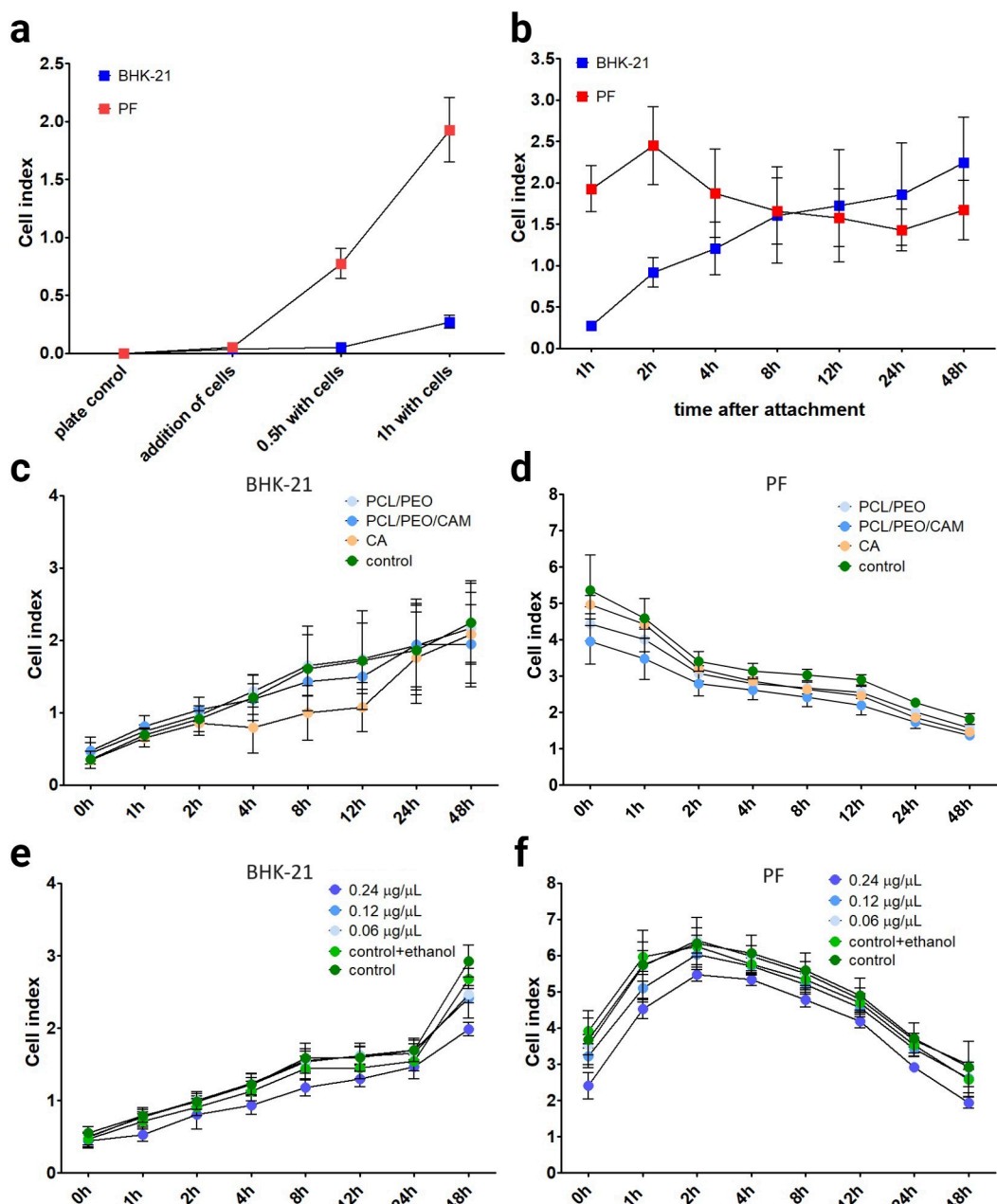

**Fig 3. The results of the indirect real-time cell analysis (RTCA) method for measuring the ES fiber mat extract and different chloramphenicol (CAM) concentrations affected viable cells.** Attachment rate to RTCA E-16 plate in baby human kidney fibroblasts (BHK-21) and primary human skin fibroblast (PF) cell lines (a). RTCA cell index development in BHK-21 and PF cell lines (b). Effect of the ES fiber mat fluid extract on BHK-21 (c) and PF (d) cells. Effect of different CAM concentrations on BHK-21 (e) and PF (f) cells. Key: PCL/PEO–fluid extract of PCL/PEO ES fiber mat; PCL/PEO/CAM–fluid extract of PCL/PEO/CAM ES fiber mat; CA: fluid extract of the cellulose acetate (CA) filter; Control—untreated cells grown at the bottom of the E-16 plate; Control + ethanol–untreated cells grown on the bottom of the E-16 plate, ethanol was added to the wells at the final concentration needed to dissolve CAM. The data represent three biological replicates from three separate experiments, shown as the mean ± SD.

Usually, such assays testing the ES fiber biocompatibility, are mostly conducted without PBS washing [16, 31–36] before assessment of cell viability because they are easier to conduct and are more rapid, but various reports in the literature describe the use of PBS wash in their

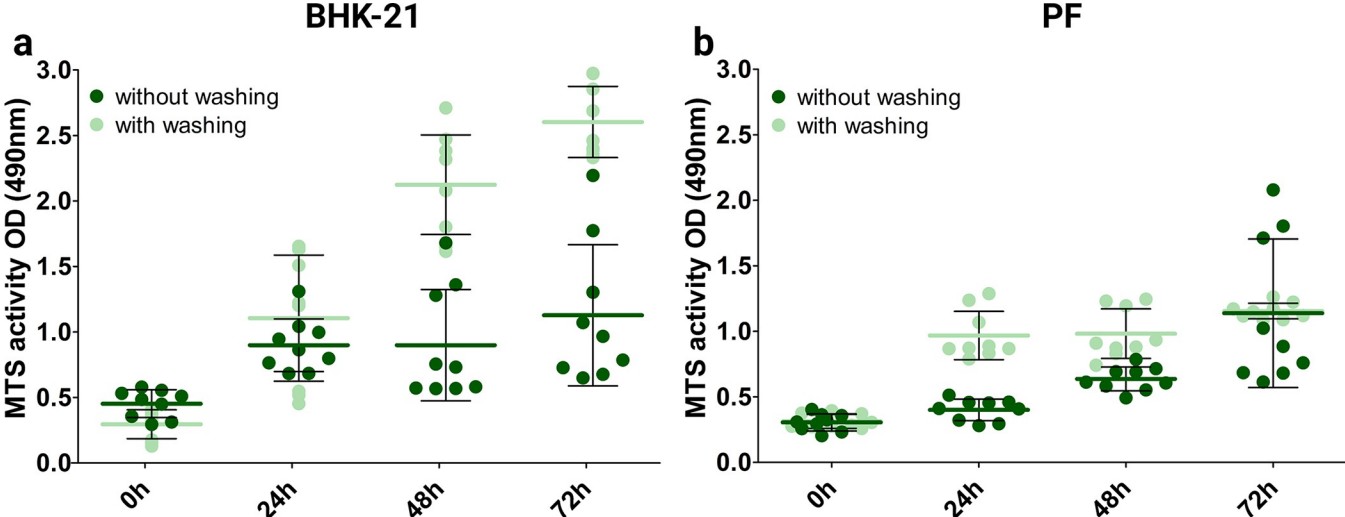

**Fig 4. Direct MTS assay results for control cells.** Comparison of MTS activity between control cells grown on the bottom of the well plate for 24 or 48 h in BHK-21 (a) and PF (b) cell lines, directly and after washing with PBS. The data represent three biological replicates from three separate experiments, shown as the mean ± SD.

experiments [15, 37, 38]. Here, we first measured MTS directly from experimental wells, and in parallel experiments, the samples were washed twice with PBS and the cell culture medium was changed before MTS measurement. As shown in Fig 4A, the MTS activity of the control well with BHK-21 cells (untreated cells at the bottom of the well plate) did not increase significantly over time in the experiment where the cells were not washed. This may give an incorrect impression of cell proliferation and, therefore, the safety of the material. When PBS washing and medium change were performed on BHK-21 cells (Fig 4A), MTS activity increased over time, indicating cell proliferation. Compared to the unwashed cells, MTS activity was at the same level at 24 h, but was much higher at 48 and 72 h (Fig 4A). This could indicate that the old growth medium (medium where cells have been grown for 72 h) limits MTS activity, because cells might not have enough nutrients to grow and therefore their metabolic activity is decreased. This phenomenon was reversed in the PF cell line (Fig 4B), where MTS activity remained at a similar level from 24 h to 72 h when cells were washed with PBS. This could indicate that PF cells did not attach as strongly as BFK 21 cells after 48 h, and a fraction of cells was lost during the washing step.

Next, we tested the direct growth of both cell lines on ES fiber mats and CA filter substrates. MTS assay was performed either directly or after washing with PBS. First, we compared the MTS activity of cells grown for 24 h directly on the test materials or in cell culture plastic control wells (Fig 5A and 5B). It was observed that cells growing on the CA filter showed lower MTS activity compared to the samples with fiber mats and the control, indicating that ES fiber materials are safe and biocompatible. However, the CA filter material showed lower MTS activity which may suggest that the material itself may bind some of the MTS, leading to erroneous conclusions about the safety of the material. Since the ES fiber materials are also known to be porous it is important to keep in mind that some level of MTS reagent adsorption and/or absorption may occur. We tested our ES fibers to rule out these possibilities. The results showed that no drastic MTS reagent binding occurred (S2 File). Visually only small colour changes of the tested ES fibers occurred (S3 File). The experiment and its results are described in the Supplementary material.

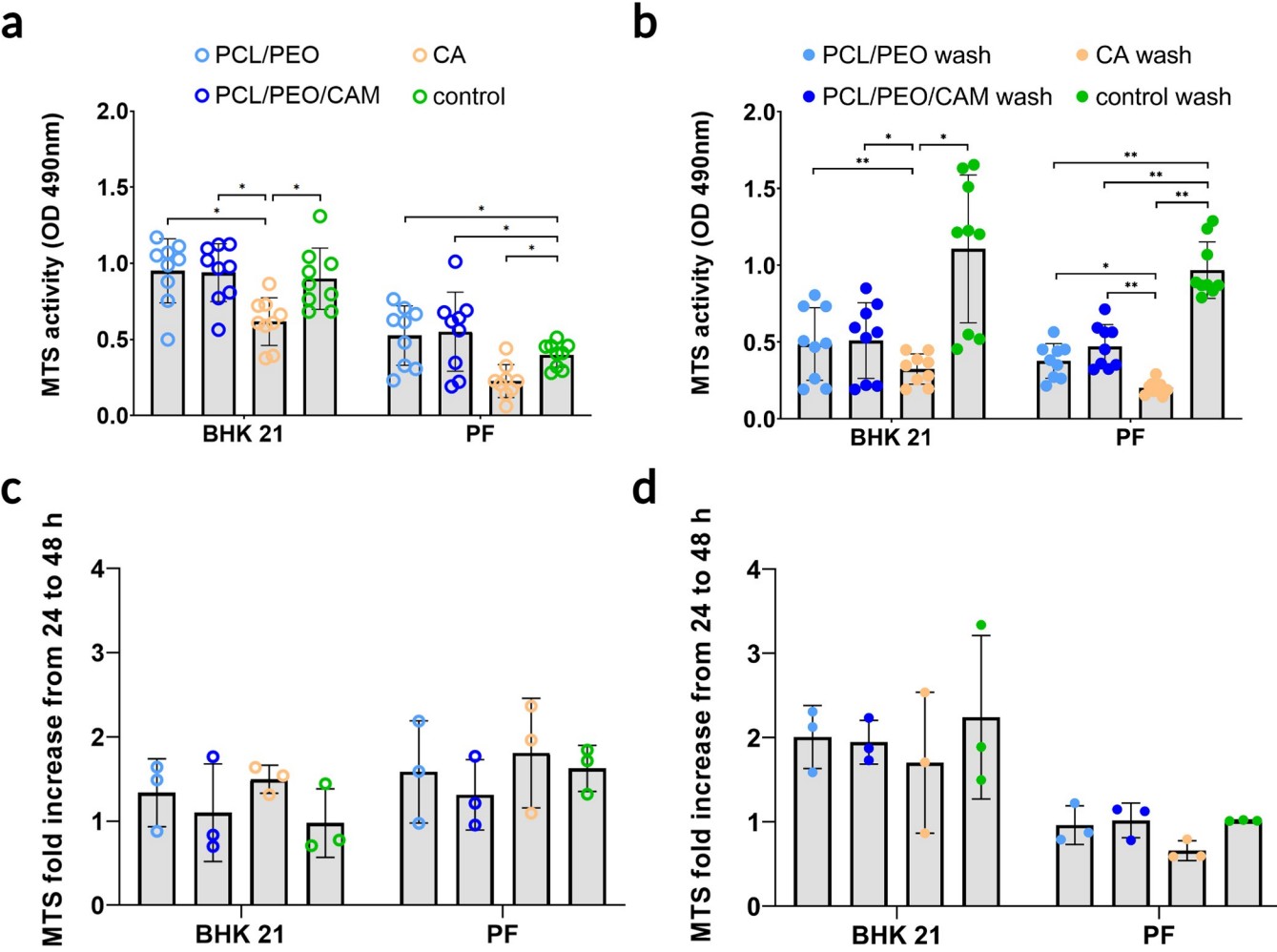

**Fig 5.** Direct MTS assay results without (a, c) and with PBS washing (b, d) at 24 h in the BHK-21 and PF cell lines. The fold increase in MTS activity was calculated based on the change from 24 to 48 h (c, d). Key: PCL/PEO—ES fiber material consisting of polycaprolactone and polyethylene oxide; PCL/PEO/CAM—ES fiber material consisting of polycaprolactone, polyethylene oxide, and chloramphenicol; CA—cellulose acetate filter used as a material control; control—cells grown at the bottom of the well plate. The data represent three biological replicates from three separate experiments and are shown as mean ± SD. * Statistically significant difference (P ≤0.05) between marked groups. **—statistically significant difference (p≤0.001) between marked groups.

As seen in Fig 5B, in all experimental conditions where cells were growing on the materials between the inserts and were washed with PBS, MTS activity was lower compared to the control conditions. The washing step before MTS measurement creates bias because fiber and filter materials with cells were transferred to new wells before MTS measurement. In addition, it was observed that some cells would end up at the bottom of the well and not on or in the ES fiber mats. This aspect needs to be considered when using substrate materials between cell culture well plate inserts (S4 and S5 Files). Therefore, it was not appropriate to compare the test group after washing only with untreated cells grown on the bottom of the plate because no cells were excluded from the control group.

To compare growth efficiency within 48 h, a fold increase in MTS activity was a more suitable estimate than comparing single time points separately (Fig 5C and 5D). The fold increase in MTS activity was calculated from the 24 h and 48 h time point MTS assay OD 490 nm

results using the following formula:

$$Fold\ increase\ in\ MTS\ activity = \frac{48\ h\ MTS\ OD}{24\ h\ MTS\ OD}$$

The obtained value provides a better understanding of cell growth over time as it describes the MTS activity increase over time. The authors used this in context as a number of estimation of cell divisions that took place between the measurement time-points. In the BHK-21 cell line, the fold increase in MTS activity for the unwashed version was lower than that of the PBS-washed version, indicating that the metabolic activity of the cells was higher for the washed version and MTS activity was not influenced by the old cell culture medium (Fig 5C and 5D). In the PF cell line, this phenomenon was reversed; the fold increase in MTS activity in the unwashed version was higher than that in the PBS-washed version (Fig 5C and 5D). This may also be due to differences in cell attachment. The results suggest that PF cells attach to the fibres weaklier than BHK-21 cells and may come off the ES fibres during PBS washing.

### Direct cell morphology assessment after cell growth on the ES fiber mat

Next, we wanted to visualise the morphology of cells grown on ES fiber mats. For this purpose, confocal microscopy together with actin filaments and cell nuclei staining was performed on the cells that were seeded at high densities and grown for 24 h on the respective substrates. Confocal microscopy images showed that when cells were grown on top of ES fibre materials, they resulted in less activated actin filaments compared to the untreated control group that was grown on top of the microscopy glass (Figs 6 and 7). From this result, we can conclude that cells change their morphology when grown in contact with ES materials compared to glass. Cell morphology and its relation to material stiffness have been previously reported in the scientific literature [39]. Stress actin filaments are less prominent in cells grown on soft surfaces [40]. Therefore, we can state that the morphological changes were not caused by the possible cytotoxicity of the ES fibers, which matches our MTS assay findings.

### Discussion

There are different physical and chemical methods to characterise ES wound dressings and scaffolds as novel biomaterials [41]. However, biological methods to test toxicity are extremely important properties, but we cannot also forget proliferation and attachment assays, inflammatory response, ROS regeneration and DNA damage assays. Standardised *in vitro* and *in vivo* evaluation assays and platforms are needed to understand the response of human cells and tissues to novel biomaterials before they can be tested in clinical trials [42].

The rationale behind the selection of the characterisation method is the intended use of these biomaterials. Various ES fiber mats have been developed for the treatment of wounds, and different cytotoxicity methods have been used for their characterization [16, 17, 23, 36, 37]. Since biorelevant testing conditions should be used for the testing of wound preparations, wound- and/or skin-mimicking conditions may vary depending on the actual mode of action of the dressing/scaffold.

In the present study, relevant biological characterisation methods, namely, cytotoxicity and biocompatibility, were evaluated. As an example, biomaterials, previously characterised as ES fiber mats for wound healing applications [18], were evaluated using different cytotoxicity assays and analytical methods. Cytotoxicity is an important property of a material, as it shows whether the material is safe or toxic to cells. This enables us to draw conclusions about its usability as a biocompatible biomaterial for various biomedical applications [20]. In cytotoxicity assays, there are several points to consider before setting up the experimental design and

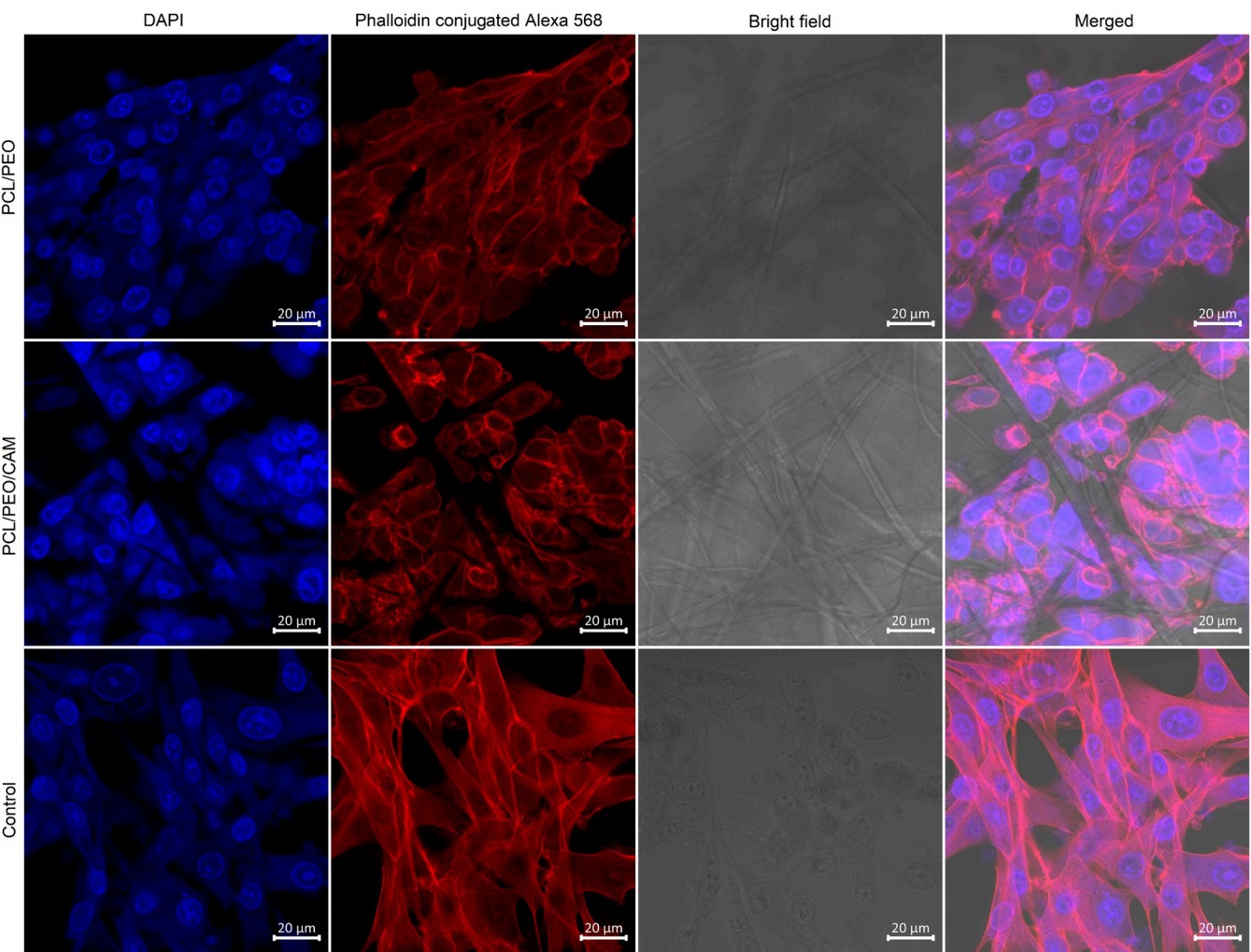

**Fig 6. Confocal microscopy images of baby hamster kidney (BHK-21) cells grown on ES fiber material for 24 h.** Key: PCL/PEO- ES fiber material consisting of polycaprolactone and polyethylene oxide; PCL/PEO/CAM—ES fiber material consisting of polycaprolactone, polyethylene oxide and chloramphenicol; and control cells were grown on microscope glass slides. Blue–DAPI, red–phalloidin conjugated Alexa 568.

conducting tests. The ISO guidelines only provide general guidance for the assay [13]. Each tested biomaterial and its specific biomedical application affect the choice of cytotoxicity assay, as well as the detailed parameters for the experiments. Hence, the use of ES fiber mats as wound dressings to cover the wound or as skin scaffolds to induce wound healing requires different testing methods.

The first choice a researcher must make before carrying out a cytotoxicity test is the choice of cell line. Currently, there are countless immortalised cell lines of both animal and human origin. It is certainly easier to carry out the experiment with an immortalised cell line because the phenotypic change over time is smaller than that in primary cells [43]. Primary cell lines are more relevant than immortalised cell lines; however, monitoring their passage number is essential. Taking all these aspects into account, we decided to perform experiments on both an immortalised animal-derived cell line and a primary patient-derived cell line isolated from humans to test PCL/PEO and PCL/PEO/CAM ES fiber mats, which are considered safe in the past [18]. Consideration should also be given to the final application of the developed biomaterial and its expected characteristics. For example, in the case of ES fibers used for wound care,

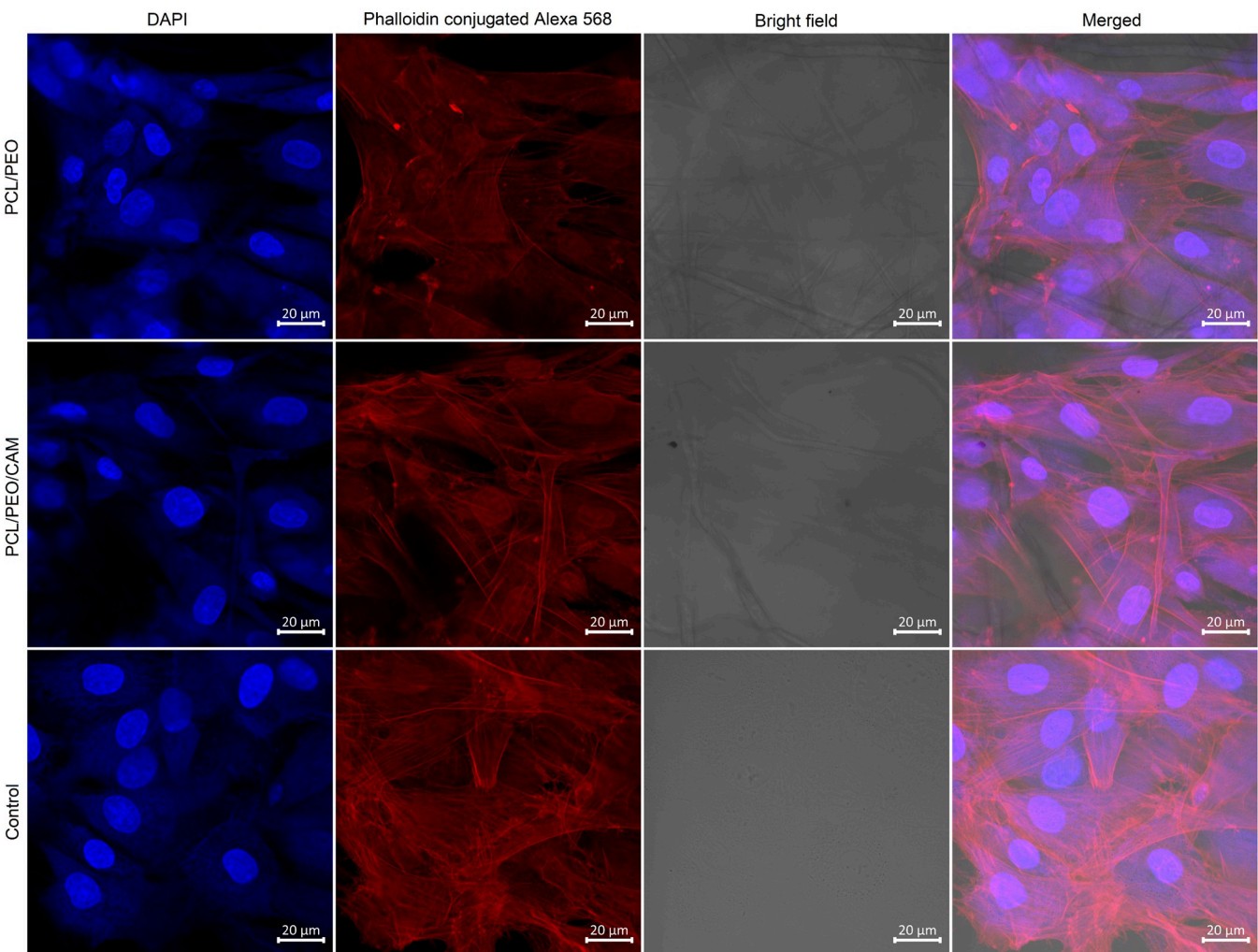

**Fig 7. Confocal microscopy images of human skin primary fibroblast (PF) cells grown on ES fiber material for 24 h.** Key: PCL/PEO- ES fiber material consisting of polycaprolactone and polyethylene oxide; PCL/PEO/CAM—ES fiber material consisting of polycaprolactone, polyethylene oxide and chloramphenicol; and control cells were grown on microscope glass slides. Blue–DAPI, red–phalloidin conjugated Alexa 568.

it is important to use human skin cells. In this study, we used primary human skin fibroblasts. Moreover, metabolically active cells are required to promote wounds to heal [44]. Although different cell lines were used to test the safety of ES materials in the extract exposure method, the protocol and conditions must be adjusted according to the properties of the cells (Fig 1). Even more challenging is the use of combinations of different cell types which is more relevant for testing wound healing preparations.

The indirect extract exposure method allowed for a simple assessment of the safety of the material in eukaryotic cells (Fig 8). It is a relatively fast method, and after 24 h, it is possible to understand whether the novel biomaterial is safe or toxic. From our previous studies, we know that CAM is released fast from the PCL/PEO/CAM ES fibers and much of the PEO is dissolved, but PCL remains the same in aqueous solution [18, 21]. Therefore, it is important to use the extract exposure method to understand the toxic effects of the released substances on eukaryotic cells. We showed that the substances released from the PCL/PEO and PCL/PEO/CAM ES fiber mats during the specified time period were safe for the cells (Figs 2 and 3). One

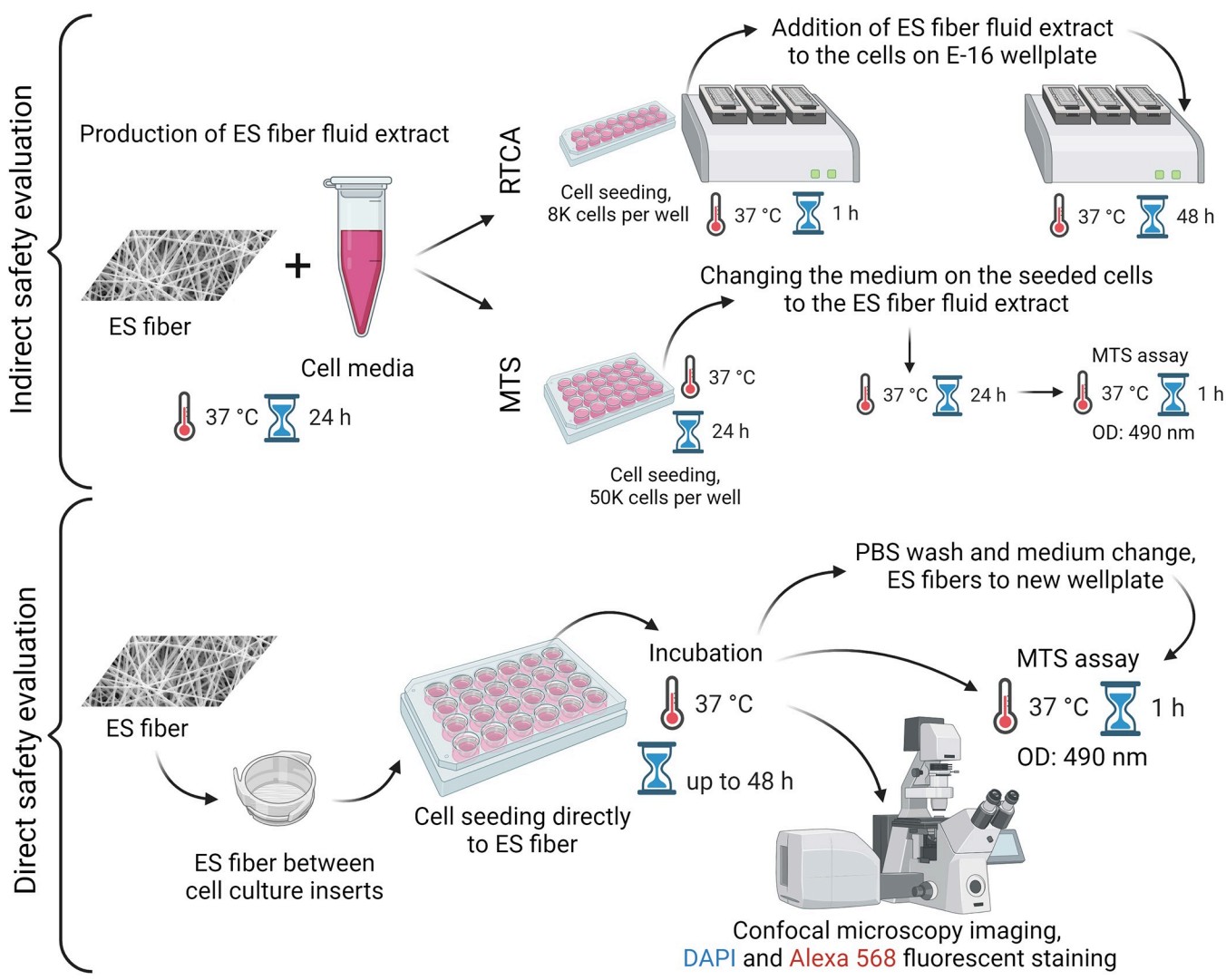

**Fig 8. Schematic of the indirect extract assay using MTS and RTCA methods and the direct contact assay with MTS method and microscopy.** Key: DAPI —4′,6-diamidino-2-phenylindole; Alexa 568 –phalloidin conjugated Alexa 568; ES—electrospun; OD–optical density.

aspect to consider is whether the method is destructive (endpoint only) or non-destructive (continuous measurement). Endpoint only measurement leaves an information gap between the two time points and is measured from different samples, but continuous monitoring gives much better understanding of cell behaviour over time. Therefore, we used continuous cell monitoring via RTCA in indirect studies to complement the endpoint results of the MTS assay. Both MTS and RTCA provided complementary results. We can see some differences in material safety in the different cell lines used in this study, because both cell lines, BHK-21 and PF, were grown in different cell culture media in addition to differences in cell lines. The traditional approach for extract exposure testing is the use of the resazurin cytotoxicity method, but the choice of the cell culture media, added serum concentration in the medium, cell seeding density and cell type can drastically influence the widely used resazurin cytotoxicity test results [45, 46]. Hence, a direct comparison between different cell lines should be performed with caution when biomaterials are tested.

One major drawback of the indirect method is that it only considers the toxic effect of the substances released from the material, and the surface of the material may also be unsuitable for cell survival. For several biomedical and pharmaceutical applications, more information is required regarding cell-fiber mat interactions to make conclusive statements about its biocompatibility and usability. It is important to understand how cells attach to the surface of the biomaterial and how this behaviour changes over time in biorelevant conditions. In this study, we demonstrated two slightly modified direct cytotoxicity measurement methods, which yielded completely different results (Fig 5). This leads us to conclude that both testing methods are needed to deeply understand the safety of the tested material, because using only one approach can lead to wrong conclusions about the safety of the material.

During direct contact testing, cells must adhere to the surface of the mats and start growing and differentiating. Often, the experiment can be designed to answer different questions; for example, testing of attachment, growth, migration, penetration, and differentiation requires different technical steps and analytical assays to be used [20]. The structure, morphology and surface properties (e.g., charge) are known to affect the behaviour of cells in contact with ES fiber mats [9, 47, 48].

Different analytical tools should be used in parallel, such as metabolic activity assays *vs*. microscopic analyses of cell morphology. Cell attachment is a dynamic process which changes over time during the course of an experiment, and cell morphology is affected by surface smoothness, stiffness, and chemical composition [39, 49]. Not only do cells change their properties, but the ES fiber mat also changes its properties in an aqueous environment. We have shown that cells change their morphology when grown on top of ES fiber mats compared to the glass used as a control (Figs 6 and 7). The size and shape of cells differ, but growth on glass is not a physiological condition for the cells, and higher activation of actin filaments was observed for the cells grown on a glass surface compared to cells grown on the fiber mat surface, presumably because of the greater need for attachment on a glass surface. Compared to plastic or glass, ES fiber material is a softer material and therefore affects the cell morphology. Soft surfaces have been shown to reduce the activity of stress actin filaments in cells [40, 50–52]. Moreover, ES fibers have a structure resembling skin´s own extracellular matrix structure and the bioactivity of the cells changes when grown on ES fiber material [53, 54]. Because fewer activated actin filaments can be seen in the confocal microscopy images (Figs 6 and 7), we can conclude that the surface of the tested ES fiber mat is more suitable for cell growth than glass. It has been previously shown that phenotypic differences are present between cells that are grown away from fibers and cells growing along fibers [55]. In our study, the structure of the ES fiber mat changed the cell morphology; however, the actual functionality of these cells requires further investigation.

One additional aspect to consider when setting up cytotoxicity testing experiments is the fact that the tested material itself can bind some amount of the testing substrate compound or product, leading the experimental results to be biased toward the cytotoxicity side [15]. Since ES fiber materials are also known to be porous, reagent binding may occur when using colorimetric assays and it could lead to incorrect interpretation of the results. This is one reason why using the MTS activity fold increase gives us a better understanding of viability and toxicity compared to the percentage calculation or comparing the raw MTS activities. In this case, it is important to consider not using too long time points because cells start forming several layers instead of a monolayer, leading to cells coming off from the material or well plate bottom. For this reason, it is necessary to consider cell seeding density, because it influences outcome of tetrazolium salt assays [46]. Moreover, metabolic activity of the cells is influenced by the excess or deficiency of nutrients [56] and larger number of cells with lesser nutrients results in decreased metabolic activity which affects the MTS assay results. Secondly, because ES material

and well-plate plastic are materials with different surface properties, it is necessary to use additional control materials in cytotoxicity experiments that resemble the test material and provide better ground for comparison. Therefore, in this study, we used CA filters as controls to compare the safety of PCL/PEO and PCL/PEO/CAM ES fiber mats. The challenge is to find suitable controls for microscopic analyses, as for real wound healing applications, CA filters or glass slides do not have the desired properties. It is suggested that controls already available and used for treatment should be selected. The major challenge is that the latter is not always possible, as innovative products differ from already available materials and direct comparison is not feasible. Although the safety and biocompatibility testing of ES fiber mats as wound dressings was performed here, it is believed that the developed testing methods can also be used for alternative biomaterial-based dosage forms, such as films and hydrogels, developed for the treatment of wounds.

## Conclusions

We developed a technical analysis protocol to evaluate the safety (cytotoxicity) of electrospun (ES) fiber mats. Setting up the experiments for safety testing starts with the cell-line choice and the appropriate design of experiments considering the actual use of the biomaterial. Fiber mat functionality testing requires more complicated experimental setups than viability testing, but with careful design, the two tests can be combined, and information can be obtained for both properties. The indirect extract method enables the investigation of the effect of chemicals released from the material on the cells, giving the first impression of the material safety, whereas direct methods provide a deeper understanding of material biocompatibility and material surface suitability for cell growth. In conclusion, there are several aspects to consider when testing ES fibers, and we suggest performing additional experiments in different ways to obtain an understanding of the tested materials from a safety perspective.

## Materials and methods

### Materials

**Materials used for electrospinning.** Chloramphenicol (CAM; PubChem CID:5959), hydrophobic carrier polymer polycaprolactone (PCL, Mw ≈80,000), and hydrophilic carrier polymer polyethylene oxide (PEO) (Mw ≈900,000) were obtained from Merck (Sigma-Aldrich, Saint Louis, Missouri, USA). Solvents methanol (gradient grade) and chloroform were also purchased from Sigma-Aldrich.

**Cell lines.** The Baby Hamster Kidney fibroblast cell line (BHK-21) and human skin primary fibroblast cell line (PF) obtained from American Type Culture Collection (ATCC) and from Estonian patient (sample was collected 19.02.2020), respectively, were used in cell studies. All methods were performed in accordance with relevant guidelines and regulations. Informed written consent was obtained from all participants prior to the study. The study was approved by the University of Tartu Ethics Committee and had an ethical committee permission (number 269/T-9). The BHK-21 cell line was chosen because it has been used previously by our research group to test ES materials safety [9, 57]. The PF cell line was used to add clinical relevance to the study.

**Control mats and buffers for *in vitro* cell experiments.** Cellulose acetate (CA) filters (GVS, USA) with a pore size of 0.45 μm were used as a control material for MTS assays.

Phosphate-buffered saline (PBS) (Corning Inc.,Corning, USA) at pH 7.4 was used in different cell culture experiments to wash the cells from the cell culture medium.

4-(2-hydroxyethyl)-1-piperazineethanesulfonic acid (HEPES) 1 M buffer (Corning Inc., Corning, USA) was added to the BHK-21 cell culture medium to stabilise the pH range.

## Methods

**Electrospinning and characterization of fiber mats.** Two different solutions were prepared for electrospinning: PCL/PEO (pure polymer solution; PCL 10% and PEO 2% (w/V) dissolved in a chloroform-methanol 3:1 (V/V) mixture) and PCL/PEO/CAM (polymer solution together with an antimicrobial drug CAM using a 4% (w/w) concentration based on the dry weight of the polymer(s). Detailed information about the solution preparation and electrospinning has been reported previously [18]. Briefly, the solutions were mixed overnight with the aid of a magnetic stirrer. Electrospinning was conducted using NanoNC electrospinning robot (South Korea). The needle used was 23G. The solution feed rate was 1 mL/h and the roller rotation speed was 25 rpm. The movement radius of the ES robot was 20–160 mm. Approximately 8 mL of the solution was electrospun to obtain fiber mats. ES fiber mats were stored at ambient conditions (temperature of $22 \pm 1°C$ and RH of $20 \pm 2\%$) until further analyses. It was verified that the prepared fiber mats reproducibly had the same morphology and physicochemical properties as those reported previously. After electrospinning, the fiber mats were kept above silica gel (0% relative humidity) under vacuum in desiccator at least for 24 h prior further analyses. Fiber mats were not additionally washed to dispose any leftover solvents from ES process and were also not additionally sanitized prior cell studies.

The morphology, diameter, and surface topography of the ES fibers were investigated by scanning electron microscopy (SEM) (Zeiss EVO 15 MA, Germany). Randomly selected areas on the fiber mats were mounted on aluminum stubs and magnetron-sputter-coated with a 3 nm platinum layer in an argon atmosphere prior to microscopy. Drug content was verified using high-performance liquid chromatography (HPLC) as shown previously [18].

**Cell culture cultivation.** Two different cell-lines were selected for the experiments to validate different safety and biocompatibility assays for novel fibrous biomaterials. BHK-21 cells were grown in Glasgow Modified Essential Medium (GMEM) (Gibco™, Thermo Fisher, USA) supplemented with 10% fetal bovine serum (FBS) (Gibco™, Thermo Fisher, USA), 2% tryptose phosphate broth (TPB) (Gibco™, Thermo Fisher, USA), 2% 1M HEPES buffer (Corning Inc., Corning, USA), 100 µg/mL penicillin, and 100 µg/mL streptomycin (Gibco™, Thermo Fisher, USA). PF were grown in Dulbeccos's Modified Eagle's Medium (DMEM) (Gibco™, Thermo Fisher, USA) supplemented with 10% FBS, 100 µg/mL penicillin, and 100 µg/mL streptomycin. Both media contained phenol red and glucose at concentration of 4.5 g/L. Cells were maintained at 37°C in a 5% $CO_2$ incubator. BHK-21 cells were used up to passage number 25, and PF cells up to passage number 12. Cells were monitored and visualised using an optical microscope (Primovert inverted microscope, Axiocam 208 colour camera, Carl Zeiss, Munich, Germany).

**MTS assays.** To determine the relative viable cell numbers and understand cell viability, proliferation, and cytotoxicity, a 3-(4,5-dimethylthiazol-2-yl)-5-(3-carboxymethoxyphenyl)-2-(4-sulfophenyl)-2H-tetrazolium (MTS) assay was used. MTS Cell Proliferation reagent (Biovison, Waltham, Boston, MA, USA) was added to the cells growing in the growth medium at a ratio of 1:10 and incubated for an additional 1 h at 37°C and 5% $CO_2$. A medium without cells was used as a control for background absorbance. Samples (200 µL) from 24-well plates were transferred to a 96-well plate, making technical duplicates for each, after which the absorbance was measured using a microplate reader (Tecan Sunrise, Tecan Group Ltd., Männedorf, Switzerland) at 490 nm.

**Real-time cell analysis.** Real-time cell analysis (RTCA) ×Celligence Instrument (ACEA Biosciences Inc., San Diego, CA, USA) was used to detect changes in cell growth, as this method allows quantitative assessment of changes in cell morphology, cell proliferation, and differentiation. RTCA Resistor plates (ACEA Biosciences Inc., San Diego, CA, USA) were

used to verify the installation and functionality of the instrument according to the manufacturer´s instructions. E-plate 16 plates (ACEA Biosciences Inc., San Diego, USA) were used for the attachment analysis and growth of the studied cells. First, 50 µL of cell medium was added to the plate wells, and the plates were inserted into the analyser for measurement. After the control step, cells were seeded onto E-16 plates. The experiment was run for the indicated times, and the impedance created by the cells was measured every 15 min. The data are shown as a cell index, which is a function of the cell number and ratio of cells at different time intervals.

**Preparation of ES fiber extracts.**   For the extract assay, ES fiber mats ($1.5 \times 1.5$ cm) were cut out and incubated in 1 mL of cell culture medium at 37˚C and 5% $CO_2$ for 24 h. All experimental steps were performed using CA filters as controls, similar to those used for ES fiber mats.

## Cytotoxicity assays for ES fibrous materials

**Indirect extract exposure assay.**   The extract exposure method is illustrated in Fig 8. MTS and RTCA assays, together with the extract exposure method, were used to test the safety of the fiber mats. For the MTS assay, cells were seeded onto 24-well plates at a density of $5 \times 10^4$ cells/well in 500 µL of cell culture medium. Cell density was determined using the trypan blue method and the cell counter Countess 3 (Invitrogen, Thermo Fisher, USA). The seeded cells were incubated at 37˚C in a 5% $CO_2$ atmosphere for 24 h before replacing the cell medium with extracts. MTS activity was measured after 24 h of incubation.

For the RTCA assay, cells were seeded onto E-16 plates at a density of $8 \times 10^3$ cells/well in 150 µL of cell medium. After 1 h of cell attachment, 150 µL of cell medium was replaced with 150 µL of ES fiber extract. The experiment was performed for 48 h, and the impedance created by the cells was measured every 15 min.

**Direct contact method.**   The direct contact method is illustrated in Fig 8. ES fiber mats in size of 1.5 x 1.5 cm were cut out and placed between CellCrown[TM] 24-wellplate inserts (Merck, Sigma-Aldrich, Saint Louis, Missouri, USA). Cells at a density of 50.000 cells per 500 µL per well were seeded onto the fiber mats in the inserts. An additional 750 µL of medium was added to the wells and incubated with the cells at 37˚C and 5% $CO_2$ for 24 h and 48 h. As a control material for MTS assays, similar to the ES fiber material, CA filters with pore size of 0.45 µm were used. All experimental steps were performed using CA filters similar to those used for ES fiber mats. In addition, cells grown at the bottom of 24-wellplates were used as controls. The MTS activity of cells was measured at the indicated time points directly from the wells. In parallel experiments, fiber mats with cells were removed from the inserts and washed twice with $1 \times$ PBS buffer. Fiber mats were placed into new 24-wellplates, and 500 µL of DMEM/nutrient mixture f-12 ham (Sigma-Aldrich, Saint Louis, Missouri, USA) without phenol red was added. Control wells containing cells were washed twice with $1 \times$ PBS, and the medium was changed before the MTS assay. Furthermore, additional MTS and light microscopy measurements were conducted to determine the number of cells located at the bottom of the wells during testing (Supporting Information, S4 and S5 Files).

**Cytotoxicity of antimicrobial agent CAM.**   The RTCA machine was set up as previously described (s.f. "Real-time cell analysis"). Cells were seeded onto E-16 plates at a density of $8 \times 10^3$ cells per 150 µL per well. After 1 h of incubation, the cells were attached to the plate and CAM solutions were added to the cells. For this, 20 µL of CAM solutions were added to obtain the desired final concentrations in the wells and three different concentrations were used: 0.24 µg/µL, 0.12 µg/µL and 0.06 µg/µL. These concentrations resembled the CAM concentrations released from the ES fiber mats. Released CAM concentration from PCL/PEO

fiber mats (0.12 μg/μL) was obtained from Preem *et al.* [18]. Untreated cells growing on the E-16 well bottom was used as controls. Untreated cells exposed to ethanol (0.1% v/v 96% ethanol in PBS) were also used as a control because ethanol was used as a solvent to dissolve CAM for the control solutions. The experiment was run for 48 h, and the impedance created by cells was measured every 15 min.

**Confocal microscopy.** Confocal microscopy (CM) LSM710 (Carl Zeiss, Munich, Germany) and Zen software (Zeiss, Germany) were used to visualise the cells on the ES fiber mats after incubation for 24 h. ES fiber mats in a size of 1.5 x 1.5 cm) were cut out and placed between CellCrown$^{TM}$ 24-wellplate inserts. Cells at a density of 2 x 10$^5$ cells in 500 μL/well were seeded onto the fiber mats in the inserts and on a round microscopy cover glass, which was used to visualise the control cells. An additional 750 μL of medium was added to the wells and incubated with cells at 37°C and 5% $CO_2$ for 24 h. After incubation, the fibers with cells were removed from the inserts and washed with 1 × PBS, after which they were fixed with 3.7% formaldehyde for 10 min at room temperature (RT). After fixation, the ES fiber mats with cells were washed with 1 × PBS. For permeabilisation, ES fiber mats with cells were treated with a 0.1% Triton-X solution for 3 min at RT. After permeabilisation, ES fiber mats were washed again with 1 × PBS. ES fiber mats with cells were then placed on a microscope slide and stained with 10 μL of staining solution containing DAPI and Phalloidin conjugated with Alexa 568 in PBS. A drop of ROTI®Mount FluorCare DAPI (Carl Roth, Karlsruhe, Germany) was then added. The sample was covered with a microscope cover class and sealed with nail polish. The samples were visualised under confocal microscope.

## Statistical analysis and graphical illustrations

The results are expressed as the arithmetic mean of three separate experiments with three biological replicates ± standard deviation (SD). Statistical analysis was performed using one-way ANOVA and post hoc pairwise t-tests with MS Excel 365 software ($p < 0.05$). In the case of multiple comparisons, Holm's method was used to adjust the p-values. Distribution of the ES fiber diameter was manually measured from 3 different SEM image by using Image J software. The number of measured fibers was N = 544 for PCL/PEO fibers and N = 488 for PCL/PEO/CAM fibers. GraphPad Prism 5 and BioRender.com were used to create the graphs and figures, respectively.

## Supporting information

**S1 File. Results of RTCA indirect ES fiber mat extract exposure assay performed on PF cells at passage numbers 12 and 5.**
(PDF)

**S2 File. Results of possible MTS reagent binding to the ES fibers in the presence of BHK-21 cells.**
(PDF)

**S3 File. Photographs of the tested ES fibers taken after each experimental step to visually detect the possible adsorption and/or absorption of the MTS reagent.**
(PDF)

**S4 File. MTS activity measured from the ES fibers and the well plate bottom after the removal of the cell culture inserts with ES fibers at 24 h and 48 h.**
(PDF)

**S5 File. Image of BHK-21 cells growing on the bottom of the plate after removal of the cell culture inserts with PCL/PEO and PCL/PEO/CAM ES fibers after 48 h of incubation.** (PDF)

**S6 File. Experimental data.** Raw data from RTCA and MTS experiments. (XLSX)

**S7 File. ZIP folder of confocal microscopy images.** Raw confocal microscopy images. (ZIP)

**S8 File. ZIP folder of SEM and light microscopy images.** Raw SEM and light microscopy images. (ZIP)

**S9 File.** (XLSX)

## Acknowledgments

Prof K. Kirsimäe is thanked for providing facilities for SEM measurements. SEM images of the CA filter were obtained at the Umeå Centre for Electron Microscopy by Dr. Linda Sandblad's research group with the help of Cheng Choo Lee.

## Author Contributions

**Conceptualization:** Karin Kogermann.

**Formal analysis:** Kaisa Põhako-Palu, Kairi Lorenz, Kelli Randmäe, Marta Putrinš.

**Funding acquisition:** Karin Kogermann.

**Investigation:** Kaisa Põhako-Palu, Kairi Lorenz, Kelli Randmäe.

**Methodology:** Kaisa Põhako-Palu, Kairi Lorenz, Kelli Randmäe, Marta Putrinš, Külli Kingo, Tanel Tenson, Karin Kogermann.

**Project administration:** Karin Kogermann.

**Resources:** Külli Kingo, Tanel Tenson, Karin Kogermann.

**Supervision:** Marta Putrinš, Külli Kingo, Tanel Tenson, Karin Kogermann.

**Visualization:** Kaisa Põhako-Palu, Kelli Randmäe, Marta Putrinš.

**Writing – original draft:** Kaisa Põhako-Palu, Karin Kogermann.

**Writing – review & editing:** Kaisa Põhako-Palu, Kairi Lorenz, Kelli Randmäe, Marta Putrinš, Külli Kingo, Tanel Tenson, Karin Kogermann.

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
