## [Decision Letter · Decision Letter 0]

20 Mar 2024

PONE-D-24-00754In vitro experimental conditions and tools can influence the safety and biocompatibility results of antimicrobial electrospun biomaterials for wound healingPLOS ONE

Dear Dr. Kogermann,

Thank you for submitting your manuscript to PLOS ONE. After careful consideration, we feel that it has merit but does not fully meet PLOS ONE’s publication criteria as it currently stands. Therefore, we invite you to submit a revised version of the manuscript that addresses the points raised during the review process.

Reviewer 1:This is a review of PONE-D-24-00754, a manuscript comparing in vitro cytotoxicity testing methods of electrospun polycaprolactone and polyethylene oxide membranes with chloramphenicol. This is a very well described, technically sound article describing different cytotoxicity testing methods. This manuscript is useful for those performing similar cytotoxicity testing as it describes a clear framework for thinking through these important in vitro experiments. I support the publication of this paper after the following comments are addressed.

Abstract

I would edit the sentence containing “true biocompatibility” to mention in vitro as one could argue that “true biocompatibility” requires animal testing.

Intro

This is only a suggestion, but the first paragraph could be shortened and collapsed into the second paragraph. It is likely that the readers of this manuscript have relevant background in biomaterials to not need the definition of biomaterials or drug delivery systems.

In the fourth paragraph, I would mention the purpose of the microscopy-based methods (LIVE/DEAD, etc) to give better context.

Methods

Information should be added about how the samples were or were not sanitized/sterilized.

Any washing of the samples after preparation, considering the solvents used, should be mentioned.

The authors should mention why these two particular cells were selected.

Results and Discussion

The figures are well-constructed and clear, congrats to the authors. However, Figures 6 and 7 should say “conjugated” – typo.

This is only a suggestion, but the authors could add information to their discussion about destructive vs. non-destructive testing methods as it relates to cytotoxicity testing.

The author should mention the degradation kinetics of these membranes as it relates to leaching monomers, etc. and differences herein between direct vs. indirect.

Another topic, only a suggestion, is discussion of differences between MTS, MTT, resazurin, etc.

Conclusion

Satisfactory, thank you authors. Reviewer 2: The manuscript titled “in vitro experimental conditions and tools can influence the safety and biocompatibility results of antimicrobial electrospun biomaterials for wound healing” is well written and highlights the necessity of investigating biocompatibility using different cell types. This is important not only for electrospun fibers but also for other nanomaterials and biomaterials for biological and biomedical applications. The manuscript does need major revision prior to acceptance.

1. The title says “antimicrobial electrospun biomaterials” while discussing the cytotoxicity of PCL/PEO electrospun fiber pristine and loaded with Chloramphenicol for wound healing applications but no antimicrobial or wound healing assessments were performed.

Introduction:

1. Line 36, “various novel biomaterials” include some examples and potential applications.

2. Line 38, What does “controversial reports can be found” mean?

3. Line 42, “The major mutual characteristic to be termed as a …………. Should be prepared for a specific purpose 1-4”, the statement is complex. Please simplify the statement.

4. Line 47, “a high surface area for efficient drug release”, high surface area provide high drug or molecule loading capacity not the release.

5. Line 52, “Providing enough oxygen and moisture to …” this does provide optimal conditions for bacterial growth as well.

6. Line 69, “with test material” not tested material.

7. Line 75-78, “after an exhaustive literature …….. (MTT, MTS) has been used.” That is true. But MTT/MTS are used for cell metabolic activity assessment and when standardized with cell count supports cell viability/toxicity assessment as well. But in general, MTT/MTS assays are used in conjunction with fluorescence or other biological assessment procedures.

Results:

1. Line 99-104, “SEM images and size analysis revealed ……….” 0.94 ± 1.56 micron for pristine and 1.03 ± 1.30 micron for CAM loaded ES fiber. That is a big standard deviation as big or bigger than measured average diameter suggesting non homogenous diameter as reported by authors. Based on this statement, how the authors can rely on their biological analysis. This large variation in ES fiber diameter questions all analyses and investigations.

2. This variation in diameter can significantly alter the physical and mechanical properties of ES fibers, not reported in manuscript.

3. Line 106, raw material’s batch variability has been reported and known but the trend can be investigated or speculated with couple of batches.

4. Line 162, as both the cell types have different size and morphology, the confluency in the well or on well surface will be different with time even after same seeding cell density because cell might have different proliferation rate.

5. Line 187, if the authors believed the different passage number can affect the behavior which is well known for primary cells. The authors should have repeated with study with same P number.

6. Line 226, metabolic activity can be a measure of comparable number cells but more metabolically active or more cells with lesser metabolic activity based on if cells like the surface or not.

7. Line 242, ES fibers are known to be porous which can adsorb MTS reagent leading to false signal.

8. Line 248, with different cells size and non-uniform ES fiber diameters, the cell distribution over the ES Fiber mat will be irregular making the any type of biological assessment a big challenge.

9. Line 253, “a fold increase in MTS activity” this is not clear.

10. Line 285, the ES fiber mat may not be cytotoxic but the biocompatibility or likeability for fiber mat for both cell types might be different, leading to modulating cell attachment, cell morphology, and or cell metabolic activity. Also, as reported by authors the ES fiber diameter was not homogenous which can tailor the ES fiber or fiber mat interaction with cells.

11. Line 300, what are different physical, mechanical, and chemical characterization techniques. Authors should have included the physical, mechanical, and chemical assessment for ES fiber pristine and CAM loaded.

12. Line 302, the biological investigation is not limited to toxicity/cytotoxicity and inflammation assay. Assays such as ROS regeneration, DNA damage, DNA/cell number quantification, proliferation assay, attachment assay, different pathway studies are equally important.

13. How about the degradation rate or degradability of ES PEO/PCL fiber with or without CAM.

Conclusion:

The ES fiber under investigation are suggested to be antimicrobial so, the antimicrobial property should be assessed with at least some of the bacterial species causing wound infection such as Pseudomonas aeruginosa, Staphylococcus aureus, Klebsiella pneumoniae, Enterococcus faecalis, and Acinetobacter baumannii. Please submit your revised manuscript by May 04 2024 11:59PM. If you will need more time than this to complete your revisions, please reply to this message or contact the journal office at plosone@plos.org. Please include the following items when submitting your revised manuscript:A rebuttal letter that responds to each point raised by the academic editor and reviewer(s). You should upload this letter as a separate file labeled 'Response to Reviewers'.A marked-up copy of your manuscript that highlights changes made to the original version. You should upload this as a separate file labeled 'Revised Manuscript with Track Changes'.An unmarked version of your revised paper without tracked changes. You should upload this as a separate file labeled 'Manuscript'.

We look forward to receiving your revised manuscript.

Kind regards,

Isha Mutreja

Academic Editor

PLOS ONE

Journal Requirements:

Estonian Research Council grant number PRG1507 (Karin Kogermann)

Reviewers' comments:

Reviewer's Responses to Questions

**Comments to the Author**

1. Is the manuscript technically sound, and do the data support the conclusions?

Reviewer #1: Yes

Reviewer #2: Partly

2. Has the statistical analysis been performed appropriately and rigorously? 

Reviewer #1: Yes

Reviewer #2: Yes

3. Have the authors made all data underlying the findings in their manuscript fully available?

Reviewer #1: Yes

Reviewer #2: Yes

4. Is the manuscript presented in an intelligible fashion and written in standard English?

Reviewer #1: Yes

Reviewer #2: Yes

5. Review Comments to the Author

Reviewer #1: This is a review of PONE-D-24-00754, a manuscript comparing in vitro cytotoxicity testing methods of electrospun polycaprolactone and polyethylene oxide membranes with chloramphenicol. This is a very well described, technically sound article describing different cytotoxicity testing methods. This manuscript is useful for those performing similar cytotoxicity testing as it describes a clear framework for thinking through these important in vitro experiments. I support the publication of this paper after the following quite minor comments are addressed.

Abstract

I would edit the sentence containing “true biocompatibility” to mention in vitro as one could argue that “true biocompatibility” requires animal testing.

Intro

This is only a suggestion, but the first paragraph could be shortened and collapsed into the second paragraph. It is likely that the readers of this manuscript have relevant background in biomaterials to not need the definition of biomaterials or drug delivery systems.

In the fourth paragraph, I would mention the purpose of the microscopy-based methods (LIVE/DEAD, etc) to give better context.

Methods

Information should be added about how the samples were or were not sanitized/sterilized.

Any washing of the samples after preparation, considering the solvents used, should be mentioned.

The authors should mention why these two particular cells were selected.

Results and Discussion

The figures are well-constructed and clear, congrats to the authors. However, Figures 6 and 7 should say “conjugated” – typo.

This is only a suggestion, but the authors could add information to their discussion about destructive vs. non-destructive testing methods as it relates to cytotoxicity testing.

The author should mention the degradation kinetics of these membranes as it relates to leaching monomers, etc. and differences herein between direct vs. indirect.

Another topic, only a suggestion, is discussion of differences between MTS, MTT, resazurin, etc.

Conclusion

Satisfactory, thank you authors.

Reviewer #2: The manuscript titled “in vitro experimental conditions and tools can influence the safety and biocompatibility results of antimicrobial electrospun biomaterials for wound healing” is well written and highlights the necessity of investigating biocompatibility using different cell types. This is important not only for electrospun fibers but also for other nanomaterials and biomaterials for biological and biomedical applications. The manuscript does need major revision prior to acceptance.

1. The title says “antimicrobial electrospun biomaterials” while discussing the cytotoxicity of PCL/PEO electrospun fiber pristine and loaded with Chloramphenicol for wound healing applications but no antimicrobial or wound healing assessments were performed.

Introduction:

1. Line 36, “various novel biomaterials” include some examples and potential applications.

2. Line 38, What does “controversial reports can be found” mean?

3. Line 42, “The major mutual characteristic to be termed as a …………. Should be prepared for a specific purpose 1-4”, the statement is complex. Please simplify the statement.

4. Line 47, “a high surface area for efficient drug release”, high surface area provide high drug or molecule loading capacity not the release.

5. Line 52, “Providing enough oxygen and moisture to …” this does provide optimal conditions for bacterial growth as well.

6. Line 69, “with test material” not tested material.

7. Line 75-78, “after an exhaustive literature …….. (MTT, MTS) has been used.” That is true. But MTT/MTS are used for cell metabolic activity assessment and when standardized with cell count supports cell viability/toxicity assessment as well. But in general, MTT/MTS assays are used in conjunction with fluorescence or other biological assessment procedures.

Results:

1. Line 99-104, “SEM images and size analysis revealed ……….” 0.94 ± 1.56 micron for pristine and 1.03 ± 1.30 micron for CAM loaded ES fiber. That is a big standard deviation as big or bigger than measured average diameter suggesting non homogenous diameter as reported by authors. Based on this statement, how the authors can rely on their biological analysis. This large variation in ES fiber diameter questions all analyses and investigations.

2. This variation in diameter can significantly alter the physical and mechanical properties of ES fibers, not reported in manuscript.

3. Line 106, raw material’s batch variability has been reported and known but the trend can be investigated or speculated with couple of batches.

4. Line 162, as both the cell types have different size and morphology, the confluency in the well or on well surface will be different with time even after same seeding cell density because cell might have different proliferation rate.

5. Line 187, if the authors believed the different passage number can affect the behavior which is well known for primary cells. The authors should have repeated with study with same P number.

6. Line 226, metabolic activity can be a measure of comparable number cells but more metabolically active or more cells with lesser metabolic activity based on if cells like the surface or not.

7. Line 242, ES fibers are known to be porous which can adsorb MTS reagent leading to false signal.

8. Line 248, with different cells size and non-uniform ES fiber diameters, the cell distribution over the ES Fiber mat will be irregular making the any type of biological assessment a big challenge.

9. Line 253, “a fold increase in MTS activity” this is not clear.

10. Line 285, the ES fiber mat may not be cytotoxic but the biocompatibility or likeability for fiber mat for both cell types might be different, leading to modulating cell attachment, cell morphology, and or cell metabolic activity. Also, as reported by authors the ES fiber diameter was not homogenous which can tailor the ES fiber or fiber mat interaction with cells.

11. Line 300, what are different physical, mechanical, and chemical characterization techniques. Authors should have included the physical, mechanical, and chemical assessment for ES fiber pristine and CAM loaded.

12. Line 302, the biological investigation is not limited to toxicity/cytotoxicity and inflammation assay. Assays such as ROS regeneration, DNA damage, DNA/cell number quantification, proliferation assay, attachment assay, different pathway studies are equally important.

13. How about the degradation rate or degradability of ES PEO/PCL fiber with or without CAM.

Conclusion:

The ES fiber under investigation are suggested to be antimicrobial so, the antimicrobial property should be assessed with at least some of the bacterial species causing wound infection such as Pseudomonas aeruginosa, Staphylococcus aureus, Klebsiella pneumoniae, Enterococcus faecalis, and Acinetobacter baumannii.

6. PLOS authors have the option to publish the peer review history of their article (what does this mean?). If published, this will include your full peer review and any attached files.

Reviewer #1: No

Reviewer #2: No

---

## [Author Response · Author response to Decision Letter 0]

3 May 2024

Answers have been provided as a separate document (together with illustrative figures)

Reviewer 1:

This is a review of PONE-D-24-00754, a manuscript comparing in vitro cytotoxicity testing methods of electrospun polycaprolactone and polyethylene oxide membranes with chloramphenicol. This is a very well described, technically sound article describing different cytotoxicity testing methods. This manuscript is useful for those performing similar cytotoxicity testing as it describes a clear framework for thinking through these important in vitro experiments. I support the publication of this paper after the following comments are addressed. 

Abstract

Q1: I would edit the sentence containing “true biocompatibility” to mention in vitro as one could argue that “true biocompatibility” requires animal testing. 

R1: We have modified the abstract of the manuscript and the word “true” was removed from the sentence to make it clearer. The text now reads:

Line 25: “We found that while extract exposure assays are suitable for the initial testing, the biocompatibility of the biomaterial is revealed in in vitro direct contact assays where cell interactions with the ES wound dressing are evaluated.” 

Intro

Q2: This is only a suggestion, but the first paragraph could be shortened and collapsed into the second paragraph. It is likely that the readers of this manuscript have relevant background in biomaterials to not need the definition of biomaterials or drug delivery systems.

R2: Thank you for your suggestion. However, after careful consideration and reading reflections from both Reviewers, we decided to leave the first paragraph on purpose longer to give readers who may not have such good background knowledge a chance to understand the topic without looking for additional external information. Indeed, for the experts in the field it is a short summary about the topic they are already familiar.

Q3: In the fourth paragraph, I would mention the purpose of the microscopy-based methods (LIVE/DEAD, etc) to give better context.

R3: In our study, microscopy was used to study cell morphology when growing in contact with ES fiber mats. The morphological changes give excellent picture about the physiological response of the cells, that is why we used confocal microscopy. In addition, when used in combination with suitable staining methods (e.g. live/dead staining) proof about cell viability, can be collected. We modified two sentences (and added relevant references), one in the fourth paragraph and the other at the end of the fifth paragraph. Sentences read now as follows: 

Line 79: “After an exhaustive literature search, we found that both extract exposure and direct contact methods have been widely used, but not always together with microscopy, which would give better understanding about cell morphological changes when in contact with material. It has been shown that fluorescent staining together with microscopy enables to enlighten the structure of cells and depending on the selected fluorophore and its sensitivity and selectivity14.”

Line 103: “The results of the cytotoxicity assay were complemented by confocal microscopy together with DAPI and Alexa 568 staining in order to illustrate the cell morphology.”

Reference: 

Alhede, M., Stavnsbjerg, C. & Bjarnsholt, T. The use of fluorescent staining techniques for microscopic investigation of polymorphonuclear leukocytes and bacteria. APMIS 126, 779–794 (2018). DOI: 10.1111/apm.12888

Methods

Q4: Information should be added about how the samples were or were not sanitized/sterilized.

Q5: Any washing of the samples after preparation, considering the solvents used, should be mentioned. 

R4 and R5: The samples were not additionally sanitized or washed after the preparations to remove any leftover solvents. However, the samples were kept in desiccator above silica gel under vacuum (0%RH) for at least 24 h prior further analyses. Additional sentences about this have been added into the Methods section: 

Line 491: “After electrospinning, the fiber mats were kept above silica gel (0% relative humidity) under vacuum in desiccator for at least 24 h prior further analyses. Fiber mats were not additionally washed to dispose any leftover solvents from ES process and were also not additionally sanitized prior cell studies.“

Q6: The authors should mention why these two particular cells were selected. 

R6: These two sentences were added to the revised manuscript: 

Line 468: “The BHK-21 cell line was chosen because it has been used previously by our research group to test ES materials safety9,57. The PF cell line was used to add clinical relevance to the study.”

References: 

Lanno, G.-M. et al. Antibacterial Porous Electrospun Fibers as Skin Scaffolds for Wound Healing Applications. ACS Omega 5, 30011–30022 (2020). DOI: 10.1021/acsomega.0c04402 

Zupančič, Š. et al. Impact of PCL nanofiber mat structural properties on hydrophilic drug release and antibacterial activity on periodontal pathogens. European Journal of Pharmaceutical Sciences 122, 347–358 (2018). DOI: 10.1016/j.ejps.2018.07.024

Results and Discussion 

Q7: The figures are well-constructed and clear, congrats to the authors. However, Figures 6 and 7 should say “conjugated” – typo. 

R7: Thank you for noticing. We have revised this and uploaded modified Figures 6 and 7. 

Q8: This is only a suggestion, but the authors could add information to their discussion about destructive vs. non-destructive testing methods as it relates to cytotoxicity testing.

R8: Thank you for your suggestion. We have added a small paragraph about this aspect to the Discussion section. The text reads as follows: 

Line 370: “One aspect to consider is whether the method is destructive (endpoint only) or non-destructive (continuous measurement). Endpoint only measurement leaves an information gap between the two time points and is measured from different samples, but continuous monitoring gives much better understanding of cell behaviour over time. Therefore, we used continuous cell monitoring via RTCA in indirect studies to complement the endpoint results of the MTS assay.” 

Q9: The author should mention the degradation kinetics of these membranes as it relates to leaching monomers, etc. and differences herein between direct vs. indirect.

R9: From our previous studies we know that CAM is quickly released from the PCL/PEO/CAM ES fibers and PEO dissolves in aqueous solutions (Preem et al. 2017 and 2019), but PCL is very durable and remains the same for longer time-period. We have added explanations about this to the revised manuscript and linked it with the choice of the indirect extract exposure method. The text reads now as follows:

Line 365: “From our previous studies, we know that CAM is released fast from the PCL/PEO/CAM ES fibers and much of the PEO is dissolved, but PCL remains the same in aqueous solution18,21. Therefore, it is important to use the extract exposure method to understand the toxic effects of the released substances on eukaryotic cells. We showed that the substances released from the PCL/PEO and PCL/PEO/CAM ES fiber mats during the specified time period were safe for the cells (Figs. 2 and 3).” 

References: 

Preem, L. et al. Interactions between Chloramphenicol, Carrier Polymers, and Bacteria-Implications for Designing Electrospun Drug Delivery Systems Countering Wound Infection. Molecular Pharmaceutics 14, 4417–4430 (2017). DOI: 10.1021/acs.molpharmaceut.7b00524

Preem et al. Monitoring of Antimicrobial Drug Chloramphenicol Release from Electrospun Nano- and Microfiber Mats using UV Imaging and Bacterial Bioreporters. Pharmaceutics 11, 487 (2019). DOI: 10.3390/pharmaceutics11090487

Q10: Another topic, only a suggestion, is discussion of differences between MTS, MTT, resazurin, etc.

R10: Thank you for the suggestion. After reflecting your comment, we decided not to deeply discuss the differences between MTS, MTT and resazurin, since this was not the main focus of this manuscript and we have referenced relevant papers discussing about these methods in our paper. 

Conclusion

Satisfactory, thank you authors.

 

Reviewer 2: 

The manuscript titled “in vitro experimental conditions and tools can influence the safety and biocompatibility results of antimicrobial electrospun biomaterials for wound healing” is well written and highlights the necessity of investigating biocompatibility using different cell types. This is important not only for electrospun fibers but also for other nanomaterials and biomaterials for biological and biomedical applications. The manuscript does need major revision prior to acceptance. 

1. The title says “antimicrobial electrospun biomaterials” while discussing the cytotoxicity of PCL/PEO electrospun fiber pristine and loaded with Chloramphenicol for wound healing applications but no antimicrobial or wound healing assessments were performed.

The aim of this study was to develop suitable methods for the in vitro safety and biocompatibility testing and highlight the aspects that may affect the results, their interpretation and what should be considered when testing such novel biomaterials. The ES fiber mats containing chloramphenicol that were used as model dressings for our safety studies have already been characterised for their antibacterial activity using different pathogenic bacteria and antimicrobial assays in our previous works (Preem et al. 2017, Preem et al. 2019, Lorenz et al. 2023). We have now added this explanation and references also to the revised manuscript:

Line 94: “We have previously published the development and full physicochemical characterisation of PCL/PEO and PCL/PEO/CAM ES fibers,18 and investigated the antimicrobial effect of the antibiotic-loaded fibers against various wound pathogens and using different antimicrobial assays18,21,22. In the present study, we tested these ES materials in two different cell lines using indirect extract exposure and direct contact cytotoxicity evaluations.”

References:

Preem, L. et al. Interactions between Chloramphenicol, Carrier Polymers, and Bacteria-Implications for Designing Electrospun Drug Delivery Systems Countering Wound Infection. Molecular Pharmaceutics 14, 4417–4430 (2017). DOI: 10.1021/acs.molpharmaceut.7b00524

Preem et al. Monitoring of Antimicrobial Drug Chloramphenicol Release from Electrospun Nano- and Microfiber Mats using UV Imaging and Bacterial Bioreporters. Pharmaceutics 11, 487 (2019). DOI: 10.3390/pharmaceutics11090487

Lorenz, K. et al. Development of In Vitro and Ex Vivo Biofilm Models for the Assessment of Antibacterial Fibrous Electrospun Wound Dressings. Mol. Pharmaceutics 20, 1230–1246 (2023). DOI: 10.1021/acs.molpharmaceut.2c00902 

Introduction:

Q1: Line 36, “various novel biomaterials” include some examples and potential applications.

R1: We have added some examples and potential applications as suggested by the Reviewer, the revised manuscript now reads:

Line 36: “Some of the examples of biomaterials are implants, heart valves, stents, lenses, prostheses, wound dressings etc1. Biomaterials are widely used in regenerative medicine, tissue engineering, orthodontics and drug delivery2.”

References:

Ratner, B. D. Biomaterials: Been There, Done That, and Evolving into the Future. Annu. Rev. Biomed. Eng. 21, 171–191 (2019). DOI: 10.1146/annurev-bioeng-062117-120940

Ajmal, S., Athar Hashmi, F. & Imran, I. Recent progress in development and applications of biomaterials. Materials Today: Proceedings 62, 385–391 (2022). DOI: 10.1016/j.matpr.2022.04.233

Q2: Line 38, What does “controversial reports can be found” mean?

R2: We have modified the text and removed this statement, and the text now reads as follows:

Line 38: The terms and definitions of “biomaterials” have changed a lot over the past decades, and nowadays artificial organs, tissue scaffolds and delivery vehicles for cells, genes and active pharmaceutical ingredients can all be classified as biomaterials3.

Reference: 

Masaeli, R., Zandsalimi, K. & Tayebi, L. Biomaterials Evaluation: Conceptual Refinements and Practical Reforms. Ther Innov Regul Sci 53, 120–127 (2019). DOI: 10.1177/2168479018774320

Q3: Line 42, “The major mutual characteristic to be termed as a …………. Should be prepared for a specific purpose 1-4”, the statement is complex. Please simplify the statement.

R3: We have simplified the definition in order to make it clearer, the statement now reads as follows: 

Line 42: “Biomaterial is a material/substance that has been engineered to become in contact with living tissues for therapeutic or diagnostic purposes.” 

Q4: Line 47, “a high surface area for efficient drug release”, high surface area provide high drug or molecule loading capacity not the release.

R4: Thank you for this remark, by accident we only had “surface area” in the text making our statement invalid. The high surface area to volume ratio is one of the key parameters that may largely affect the drug release properties. That is the reason why “nano” systems provide advantages over the larger particles (or fibers with larger diameters) and these nanotechnological biomaterials enable to increase the dissolution rate and release of poorly water-soluble drugs from ES fibers. We have revised the manuscript as follows:

Line 47: “ES mats have several prominent features, such as a porous structure, the abilities to incorporate drugs, including antibiotics, into fiber structures, and a high surface area to volume ratio for efficient drug release8.”

Reference: 

Luraghi, A., Peri, F. & Moroni, L. Electrospinning for drug delivery applications: A review. Journal of Controlled Release 334, 463–484 (2021). DOI: 10.1016/j.jconrel.2021.03.033 

Q5. Line 52, “Providing enough oxygen and moisture to …” this does provide optimal conditions for bacterial growth as well.

R5: Indeed, these are also very good conditions for bacteria to grow, but it is well known that bacteria and eukaryotic cells prefer to live in somewhat similar environments. Properties such as good moisture retention, air permeability and a fibrous surface are precisely what is sought in wound dressings. Unfortunately, there is no way to avoid the fact that in addition to supporting the wound healing, wound dressings also have bacterial growth-promoting properties. For this reason, we are developing wound dressing materials in which we can incorporate antibiotics and other antimicrobial agents that are safe for eukaryotic cells and support the wound healing but inhibit the bacterial growth.

We added the following sentence to the revised manuscript:

Line 55: “More importantly, the presence of antimicrobial agent in ES wound mats is relevant as this inhibits the growth of bacteria which is also more favoured in these mentioned environmental conditions developed.”

Q6: Line 69, “with test material” not tested material. 

R6: We replaced the word “tested” with “test” (line 75).

Q7: Line 75-78, “after an exhaustive literature …….. (MTT, MTS) has been used.” That is true. But MTT/MTS are used for cell metabolic activity assessment and when standardized with cell count supports cell viability/toxicity assessment as well. But in general, MTT/MTS assays are used in conjunction with fluorescence or other biological assessment procedures.

R7: We agree that it is beneficial to use MTT/MTS assays together with other suitable methods to gather more relevant information. Therefore, we used RTCA and microscopy with fluorescent staining, in addition to the MTS assay. However, the ISO 10993-5 standard, which is the basis of safety testing for novel materials, recommends choosing and using only one method; MTT is one such method in the ISO standard list. We agree completely with the reviewer that different methods have to be combined and this was one of the reasons for preparing the current manuscript.

Results:

Q1. Line 99-104, “SEM images and size analysis revealed ……….” 0.94 ± 1.56 micron for pristine and 1.03 ± 1.30 micron for CAM loaded ES fiber. That is a big standard deviation as big or bigger than measured average diameter suggesting non homogenous diameter as reported by authors. Based on this statement, how the authors can rely

---

## [Decision Letter · Decision Letter 1]

27 May 2024

In vitro experimental conditions and tools can influence the safety and biocompatibility results of antimicrobial electrospun biomaterials for wound healing

PONE-D-24-00754R1

Dear Dr. Kogermann,

We’re pleased to inform you that your manuscript has been judged scientifically suitable for publication and will be formally accepted for publication once it meets all outstanding technical requirements.

Kind regards,

Isha Mutreja

Academic Editor

PLOS ONE

Reviewers' comments:

Reviewer's Responses to Questions

**Comments to the Author**

1. If the authors have adequately addressed your comments raised in a previous round of review and you feel that this manuscript is now acceptable for publication, you may indicate that here to bypass the “Comments to the Author” section, enter your conflict of interest statement in the “Confidential to Editor” section, and submit your "Accept" recommendation.

Reviewer #1: All comments have been addressed

2. Is the manuscript technically sound, and do the data support the conclusions?

Reviewer #1: Yes

3. Has the statistical analysis been performed appropriately and rigorously? 

Reviewer #1: Yes

4. Have the authors made all data underlying the findings in their manuscript fully available?

Reviewer #1: Yes

5. Is the manuscript presented in an intelligible fashion and written in standard English?

Reviewer #1: Yes

6. Review Comments to the Author

Reviewer #1: The authors have responded in details to my questions and suggestions for improvement. I have gone through in detail the response to me and have skimmed the other reviewer response. I have no further comments and want to to congratulate the authors on a very clear and technically sound paper.

7. PLOS authors have the option to publish the peer review history of their article (what does this mean?). If published, this will include your full peer review and any attached files.

Reviewer #1: No

---

## [Editor Report · Acceptance letter]

30 May 2024

PONE-D-24-00754R1 

PLOS ONE

Dear Dr. Kogermann, 

I'm pleased to inform you that your manuscript has been deemed suitable for publication in PLOS ONE. Congratulations! Your manuscript is now being handed over to our production team.

Kind regards, 

on behalf of

Dr. Isha Mutreja 

Academic Editor

PLOS ONE